# Lateral Humeral Condyle Fractures in Pediatric Patients

**DOI:** 10.3390/children10061033

**Published:** 2023-06-08

**Authors:** Tim F. F. Saris, Denise Eygendaal, Bertram The, Joost W. Colaris, Christiaan J. A. van Bergen

**Affiliations:** 1Department of Orthopedic Surgery, Amphia Hospital, 4818 CK Breda, The Netherlands; 2Department of Orthopaedics and Sports Medicine, Erasmus University Medical Center—Sophia Children’s Hospital, 3015 GD Rotterdam, The Netherlands

**Keywords:** lateral humeral condyle, fracture, children, diagnosis, treatment, surgery

## Abstract

Lateral humeral condyle fractures are frequently seen in pediatric patients and have a high risk of unfavorable outcomes. A fall on the outstretched arm with supination of the forearm is the most common trauma mechanism. A physical examination combined with additional imaging will confirm the diagnosis. Several classifications have been described to categorize these fractures based on location and comminution. Treatment options depend on the severity of the fracture and consist of immobilization in a cast, closed reduction with percutaneous fixation, and open reduction with fixation. These fractures can lead to notable complications such as lateral condyle overgrowth, surgical site infection, pin tract infections, stiffness resulting in decreased range of motion, cubitus valgus deformities, ‘fishtail’ deformities, malunion, non-union, avascular necrosis, and premature epiphyseal fusion. Adequate follow-up is therefore warranted.

## 1. Introduction

The occurrence rate of any fracture during childhood (0–17 years) is between 12–34% for boys and 6–34% for girls [1,2]. Elbow fractures account for 28.4% of this number [3]. Lateral humeral condyle fractures in children are the second most frequent type of elbow fracture [4,5,6,7]. In most cases, standard radiographs will visualize the fracture. However, in 5.2–16.6% of all cases, nondisplaced fractures are overlooked on conventional radiographs, which could result in an unfavorable outcome or long-term complication [8,9,10]. Therefore, early recognition and adequate treatment are necessary to optimize outcomes [8]. The objective of this narrative review is to provide an overview of the epidemiology, anatomy, diagnosis, treatment options, and complications of pediatric lateral humeral condyle fractures based on the most recent literature.

## 2. Epidemiology

Lateral humeral condyle fractures account for 9.6–22.3% of all elbow fractures [1,2,4,5,6,7,11,12,13,14]. The majority (63–67.4%) of pediatric patients are male [8,9,13,14,15]. Most lateral humeral condyle fractures are seen between the ages of four and ten, but cases as early as 1.9 years have been reported [5,6,7,8,9,11,13,16,17]. Lateral humeral condyle fractures are typically a result of playground activities (53.7%) and/or sports (49.6%) [6,8,9], and as a consequence, they do occur more than twice as much during the summer period [5]. The handedness of the patient has an effect on fracture occurrence in the elbow: the non-dominant side fractures more often than the dominant side [18]. A slight majority (51.5%) of lateral humeral condyle fractures are nondisplaced fractures or show minor displacement (<2 mm) in plain radiography [8,9].

### 2.1. Anatomy

The capitulum of the humerus and the lateral condyle demarcate the anatomical region of the lateral side of the elbow [19]. The ossification center of the capitulum develops first at the age of one, and the ossification center of the lateral condyle develops last at the age range between eight and thirteen [20]. The ossification process toward the total osseous fusion of the lateral side of the elbow is completed between the ages of twelve and fourteen. The blood supply for the tissue on the lateral side originates posterior from the branched variation of the radial collateral artery over the lateral condyle and the brachial artery between the capitulum and the humeral trochlea. The cephalic vein and accompanying lymph vessels traverse the capitulum toward the radial head as seen in Figure 1. The radial fossa is a slight anatomical depression of the humerus just above the capitulum, where the brachial artery (Figure 1) and radial nerve track are located.

### 2.2. Trauma Mechanism and Associated Injuries

The most common trauma mechanism resulting in a lateral humeral condyle fracture is a fall on an outstretched arm with the wrist in full supination [16]. Another common trauma mechanism is a direct hit to the lateral side of the elbow [6,9]. These mechanisms result in either a varus injury with an avulsion fracture of the lateral condyle and possible concomitant fracture of the capitulum or direct impact of the radial head into the lateral condyle resulting in an impaction fracture.

### 2.3. Classification of Fractures

Several classifications have been described to categorize the different fracture patterns. Most of these classifications use anatomical landmarks of the elbow and the amount of displacement to determine the severity of the fracture. The most commonly used classifications, presented in Table 1, in historical chronology are Milch (1956), Jakobs (1975), Finnbogason (1995), Weiss (2009), and Song (2010).

The Milch Classification [16], designed in 1956, is a classification that uses the anatomical regions within the elbow to define the different types of lateral humeral condyle fractures. Milch type 1, with an occurrence of 11.4–50.7% [8,9,21], is a fracture through the capitulum humeri with or without the involvement of the lateral side of the trochlear groove. This fracture type occurs after an axially loaded trauma to the elbow. Milch type 2, with an occurrence of 49.3–88.6% [8,9,21], is a fracture of the lateral condyle and part of the medial trochlear groove in the capitulum humeri. This fracture results from ulnar outward rotation with lateral displacement of the distal fragment.

The Jakobs Classification [21], designed in 1975, is a classification that differentiates lateral humeral condyle fractures based on the anatomical region and the extent of displacement of the fractures seen on conventional radiographs. Jakobs type 1, with an occurrence of 8–51.5% [8,9], is a fracture with minimal displacement (<2 mm) and without discontinuation of the articular surface of the capitulum/trochlea (incomplete fracture). Jakobs type 2, with an occurrence of 29.9–65% [8,9], is a fracture with minimal displacement (between 2 and 4 mm) and a discontinuation of the articular surface. Jakobs type 3, with an occurrence of 18.7–27% [8,9], is a displaced fracture with discontinuation of the articular surface and displacement measuring > 4 mm.

The Finnbogason classification [22], designed in 1995, is a modified version of the Jakobs classification based on anatomy and displacement of the articular surface as seen using conventional radiographs. Type A is a lateral humeral condyle fracture with a small gap without displacement or discontinuation of the articular surface. Type B is a fracture with a small gap without displacement but with discontinuation of the articular surface. Type C is a fracture with a considerable gap, displacement, and discontinuation of the articular surface.

The Weiss classification [23], designed in 2009, is a modified version of the Milch classification. Weiss et al. found that the Milch classification inadequately divided fractures based on their potential treatment options. More focus was placed on articular congruity and the potential to guide within the treatment options. Weiss type 1 is a lateral humeral condyle fracture with an intact articular surface and displacement of <2 mm. Weiss type 2 is a fracture with an intact articular surface and displacement of >2 mm. Weiss type 3 is a fracture with the incongruity of the articular surface and displacement of >2 mm.

**Table 1 children-10-01033-t001:** Visual summarization of the most commonly used classifications for lateral humeral condyle fractures in pediatric patients in historical chronology. The Finnbogason classification type is described with alphabetical numeration within brackets.

Classification	Type 1 (A)	Type 2 (B)	Type 3 (C)	Type 4	Type 5
Milch. [16]	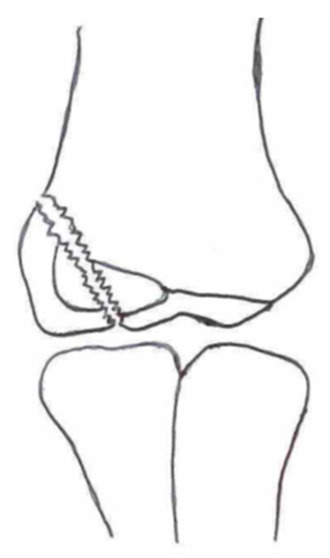	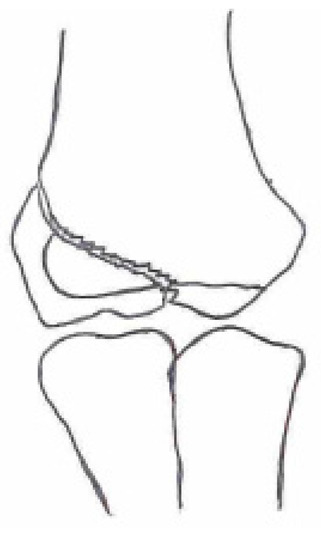			
Jacobs et al. [21]	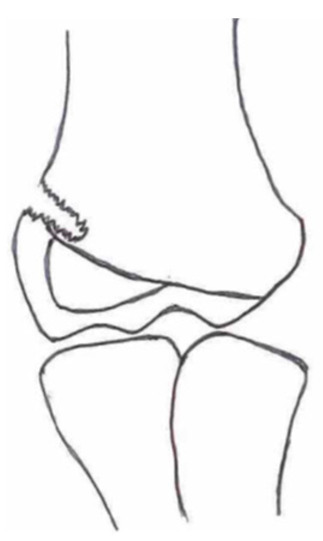	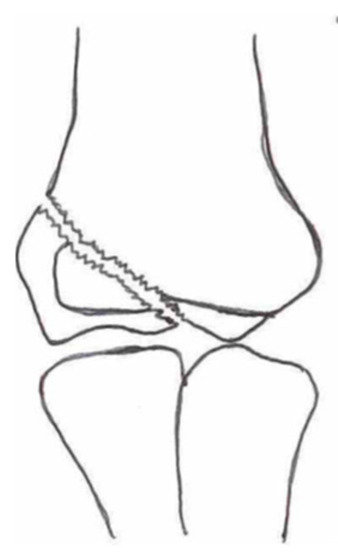	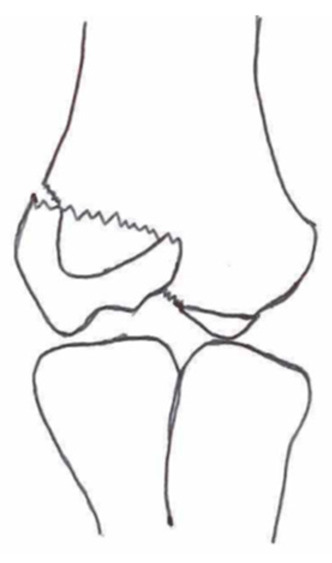		
Finnbogason et al. [22]	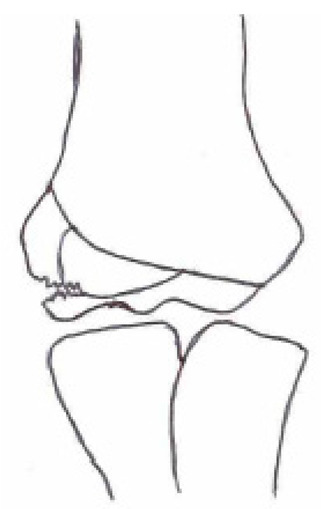	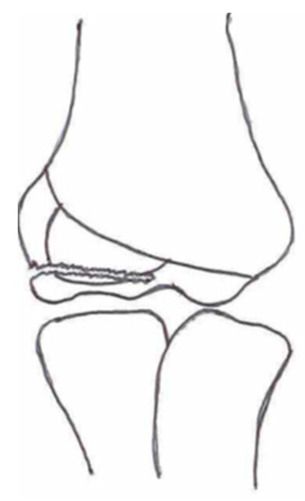	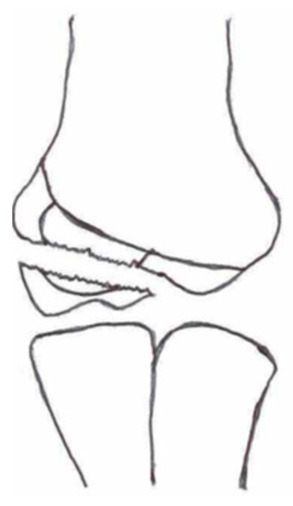		
Weiss et al. [23]	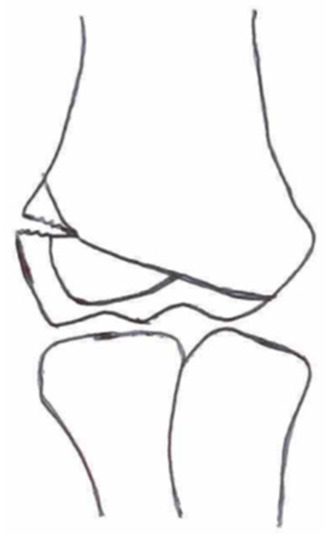	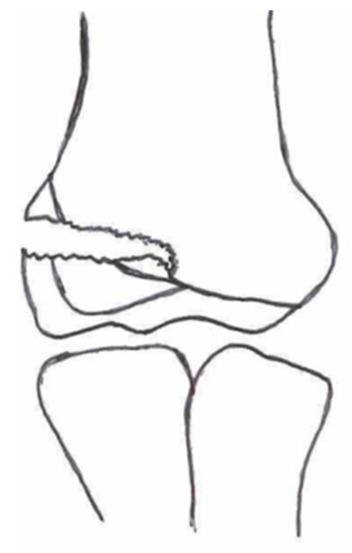	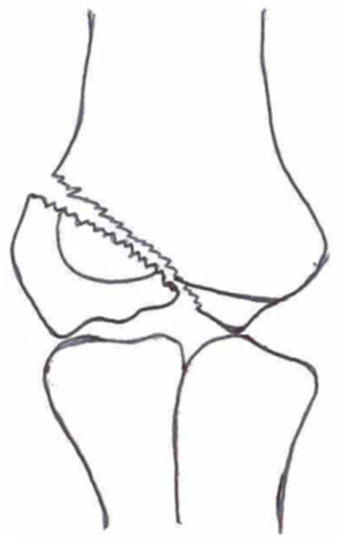		
Song et al. [4]	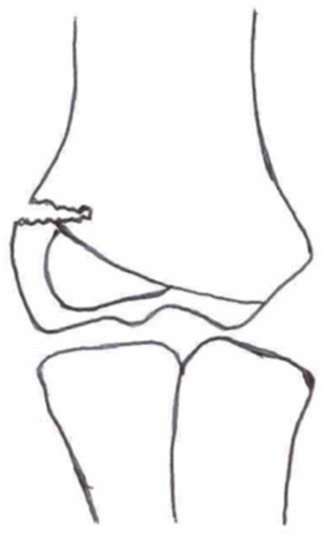	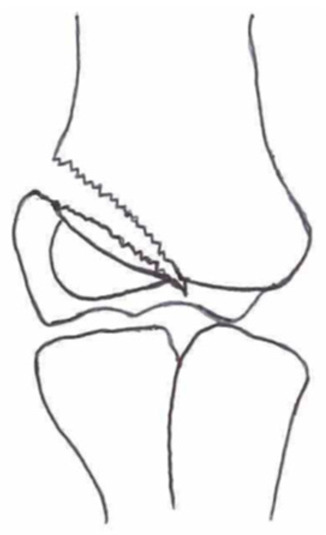	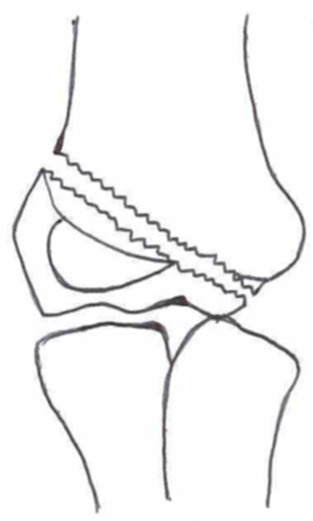	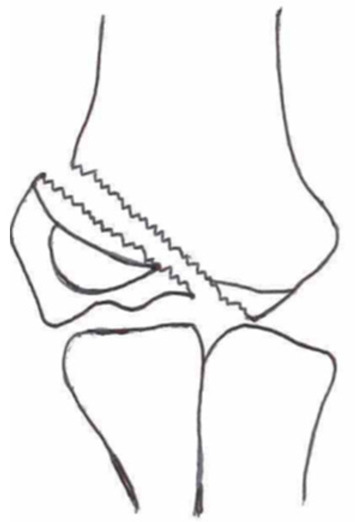	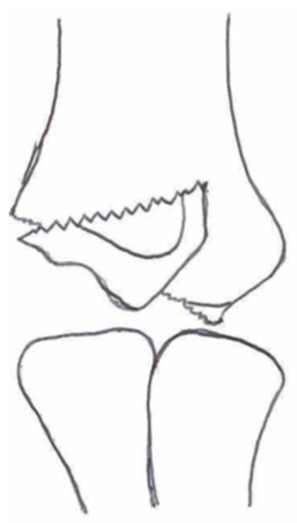

The Song classification [4] is the most recent updated classification, focusing on conventional radiographs, especially the internal oblique view. Song type 1 is a stable lateral humeral condyle fracture confined to the metaphyseal bone on all radiographical views with a displacement of <2 mm. Song type 2 is a potentially unstable fracture with <2 mm displacement through the cartilaginous layer but showing no intra-articular fracture. Song type 3 is an unstable intra-articular fracture with a displacement of <2 mm. Song type 4 is an unstable intra-articular fracture with a lateral displacement of >2 mm. Finally, Song type 5 is an unstable intra-articular fracture with a rotational displacement of >2 mm.

Rarely is the Milch classification used to identify lateral humeral condyle fractures, mainly because it focuses on the anatomical position of the fracture and does not direct the user toward treatment options [24,25]. The Song classification compiles the positives taken from Weiss and Jakobs’s classifications. The Song classification divides fractures based on the indication criteria for surgical interventions or non-surgical treatment. Ramo et al. have extensively tested and validated the interobserver reliability of the Song classification and its ability to determine the correct treatment option accurately [26]. The Song classification might therefore be preferred.

### 2.4. Diagnosis

Children with an elbow injury suspected of an elbow fracture usually present with an adequate trauma mechanism, localized pain, possible deformity, swelling, limitation in elbow range of motion, and inability to use the injured arm. Although standard radiographs in the anterior–posterior and lateral direction can confirm the diagnosis, additional diagnostic imaging, such as an internal oblique view, X-rays of the contralateral elbow, computed tomography (CT), and magnetic resonance imaging, are helpful in selected cases. A physician should also assess the possibility of child abuse as a trauma mechanism for the elbow fracture and look for potential additional injuries.

#### 2.4.1. Medical History and Physical Examination

A patient’s medical history should include detailed information on identifiable risk factors for an elbow fracture. These risk factors include but are not limited to age, gender, physical activities leading to trauma, trauma mechanism, external visual deformity, and sensorimotor function. During a physical examination, visible deformity of the elbow joint and localized ecchymosis due to tearing blood vessels or hematoma from articular capsular bleeding are solid indicators of a potential fracture. Other strong indicators are a reduced range of motion or inability to move, sharp pain, and crepitations felt during palpation [27]. A full assessment of the forearm’s sensory and motor function and a vascular examination of the radial and ulnar arteries will complete the physical examination.

#### 2.4.2. Diagnostic Imaging

If the suspicion of a lateral humeral condyle fracture or other elbow fracture is raised, diagnostic imaging will be the next step. First, conventional radiographs will be taken from two angles: an anterior–posterior view with the arm in supination and as much extension as possible, and a lateral view with the elbow in a 90-degree flexion and neutral rotation. Recent studies have shown that fractures and/or displacement of the lateral humeral condyle are often missed with the conventional approach [10,28,29]. Therefore, it is suggested to perform additional imaging by taking a conventional radiograph of the uninjured elbow or by taking an internal oblique view (anterior–posterior position with hand in full pronation) [10]. Furthermore, a fat-pad sign is a radiographic finding that suggests hydrops or hematoma of the joint and, thus, a possible fracture. A radiograph with a fat-pad sign and no evident fracture is usually considered as no fracture [30,31]. However, recent studies showed that 44.6% (confidence interval: 30.4% to 59.7%) of children with a positive fat-pad sign and no evident fracture visible on conventional radiographs have an occult fracture [31]. In total, 14% of these occult fractures consisted of lateral humeral condyle fractures [31]. Therefore, when in doubt, it is advised to perform additional CT-imaging to appreciate the extent of injury to the elbow joint fully, define the proper classification, and execute the suitable treatment method [32]. Historically, elbow surgeons have used 2 mm and 4 mm as the cut-off values to determine conservative or surgical treatment. The literature does not accurately describe the origin of these margins and whether or not these margins are optimal. Nevertheless, the 2 mm and 4 mm values are now common practice. The distance is measured between the most lateral gap between the fracture site and the fractured bone piece. It is the biggest measurable distance between bone and fractured bone. This measurement is susceptible to measurement errors due to the thick cartilaginous layer surrounding the articular surface of the bone in children and the inability to visualize the cartilage on fluoroscopy and/or radiography. If a fracture shows displacement and rotation, the protrusion of a thick cartilage layer between the fracture site and the bone piece could overestimate the distance of the gap and have significant implications for the correct treatment option.

## 3. Treatment Options

Non-operative treatment is the preferred option for fractures with a minor (<2 mm) displacement and no other additional injuries [33]. Closely regulated follow-up is mandatory to rule out secondary displacement in the cast. Follow-up should be performed within one week after trauma in the outpatient clinic and should include conventional X-rays in AP and lateral and oblique views. If the fracture is >2 mm displaced, with a disruption of the articular surface, reduction and fixation of the fracture are recommended [33,34]. The authors of each classification system for lateral humeral condyle fractures have made recommendations concerning treatment options based on the severity of the fractures; these recommendations can be found in Table 2. In cases with a successful fracture union and without complications, the success rate for non-operative and operative lateral humeral condyle fracture, as described in meta-analyses, is between 89.8–91.5% [8,33,35,36]. A delayed diagnosis of these fractures (>3 weeks after injury) should initially be given based on the time between the injury and presentation and the amount of displacement, in accordance with the above-described options [37,38]. Unfortunately, a malunion or nonunion of these fractures with a delayed diagnosis is quite common after 3 months [37]. Treatment of these complications requires a different approach depending on displacement, elbow alignment, and a stable condylar fragment (see Section 3.2.1 and Section 3.2.2).

### 3.1. Non-Surgical/Operative Treatment Options: Plastered Cast Therapy

The non-operative treatment option for lateral humeral condyle fractures in children is an above-elbow cast. This applies to a fracture with no displacement, an intact articular surface, and no additional injury [11,16,17,21,33,34]. The elbow should be positioned in a 90-degree flexion, and the wrist and hands should be in a neutral position (Figure 2). Patients will return after 4 weeks for cast removal if non-operative treatment shows no secondary fracture displacement on the X-ray in a long arm cast within the first week after injury [8,10,11,33,34,39,40]. If the physician, after removal, doubts whether cast therapy for 4 weeks has been enough, an additional X-ray should be made. If the X-ray shows no callus around the fracture, treatment with plastered cast therapy should be continued for another 2 weeks [8,10,11,33,34,39,40]. Secondary fracture displacement, which warrants an operative treatment [34,41] (unstable fracture, see classifications Table 1), occurs most frequently between three and seven days after injury [33,41]. Secondary displacement of lateral humeral condyle fractures treated with a cast occurs in 4.8–29.4% of all pediatric cases [4,10,33,34,35,39,41,42].

Plastered cast therapy for a patient with a malunion or nonunion after >3 months and after an initial delayed diagnosis is only viable if the displacement is less than 5 mm, shows a stable condylar fragment, and shows evidence of bony bridging on a CT scan [37].

### 3.2. Surgical/Operative Treatment Options

#### 3.2.1. Closed Reduction and Internal Fixation

The minimally invasive surgical technique to reduce and stabilize the lateral humeral condyle fracture is called closed reduction and internal fixation (CRIF), or closed reduction and percutaneous pinning (CRPP). This technique is generally used for unstable/displaced fractures with 2 mm–4 mm displacement [43]. Most fractures that do not show signs of rotation of the fragment and/or additional fractures of the elbow are treated with CRPP. Fracture reduction through CRPP is achieved by flexing the elbow and supinating the wrist while applying pressure to the lateral side of the elbow. Simultaneous imaging should be performed to deduce the effects of the closed reduction. Successful reduction shows an anatomical articular surface during imaging. Fluoroscopy and ultrasound-guided reduction are suitable options to provide basic imaging during surgery.

Ultrasound-assisted reduction creates the opportunity to provide basic imaging of good quality while negating the negative effects of fluoroscopy radiation [44]. The image quality and ability to perform the surgery is linked to the imaging capabilities of the surgeon when using ultrasound. Ultrasound-assisted closed reduction is a relatively new technique with a learning curve for the surgeon. Nevertheless, recent results show comparable complication rates to closed reduction with fluoroscopy and/or ORIF [44].

Through simultaneous fluoroscopy, one can deduce the effects of the reduction. However, the diagnostic accuracy of joint reduction appreciated on two-dimensional fluoroscopy used in the operating theater shows inferior results compared to a CT scan [45]. The subjective image quality of fluoroscopy is the main contributing factor toward inferior diagnostic accuracy. The imaging quality is affected by the degree of image focus achieved during surgery. The most notable factor which directly impacts the quality of the image is the presence of osteosynthesis material, which results in scattering and artifacts. A secondary factor influencing the image quality is the relative thickness of the cartilage, which is more prominent in children than adults, compared to bone thickness. The cartilage and articular surface of the elbow in children are not as visible using fluoroscopy as they would be through arthrotomy since fluoroscopy does not show cartilaginous tissue as clearly as bone. Considering these factors, and combined with the over-estimation in the measurement of displacement seen in radiography prior to surgery, it is best to visualize a joint reduction through arthrography, arthroscopy, or arthrotomy.

Next, the surgeon performs a percutaneous fixation of the reduced fracture by placing two smooth Kirschner wires perpendicular to the fracture line. Crossed Kirschner wires may reduce fracture stability [46]. A third Kirschner wire can be placed through the condyles, parallel to the joint, to increase fracture stability and minimize rotation. Kirschner wires can be buried underneath the skin or exposed for easy removal. Both methods show similarly low complication rates, low infection rates, and high successful union rates. Kirschner wires are left in place for 4 weeks after surgery. In addition, the patient receives a long arm cast with elbow back slab support for 4 weeks.

CRPP for a patient with a malunion or nonunion after >3 months and after an initial delayed diagnosis is only viable if the displacement is less than 5 mm, shows an unstable condylar fragment, and shows no evidence of bony bridging on a CT scan [37].

#### 3.2.2. Open Reduction and Internal Fixation

Open reduction and internal fixation (Figure 3) is the preferred surgical treatment option for a fracture showing more than 4-mm displacement and/or rotation of the fragment. It is also the next step-up surgical option when CRPP fails to reduce the fracture to an anatomic situation. A small incision is made on the anterolateral side of the elbow. Subsequent careful dissection of the subcutaneous tissue, fascia, and articular capsule is performed. The malrotation of the fracture’s fragment and size warrants careful dissection not to disrupt the distal humerus’s blood supply and/or harm the radial nerve bundle. Like with CRPP, the surgeon will fix the fracture by placing two smooth Kirschner wires perpendicular to the fracture line. The postoperative treatment is similar to that of the CRPP. Surgeons can opt for screw fixation with a small AO bone screw combined with K-wires for rotational stability of the fragment. However, studies show screw fixation results in comparable quality of life and range of motion postoperatively while having disadvantages such as second surgery to remove the screw, impairment of the range of motion, delayed union, and wound infections [36,43].

ORIF for a patient with a malunion or nonunion after >3 months and after an initial delayed diagnosis is viable if the displacement is greater than 5 mm or is less than 5 mm with a normal elbow alignment [37,38]. If the patient has an elbow malalignment, a corrective osteotomy with simultaneous anterior transposition of the ulnar nerves can be performed [37,38].

## 4. Complications

Complications can occur during and after treatment. One in ten patients with lateral humeral condyle fractures has severe complications of the fracture and/or treatment [8,33,36]. Unsuccessful treatment of this complicated and menacing fracture may lead to a long-term loss in quality of life for pediatric patients. To minimize the risk of complications during treatment, the attending physician benefits from consulting an experienced pediatric elbow surgeon when discussing treatment options. The most notable complications are lateral condyle overgrowth, surgical site infection, pin tract infections, stiffness resulting in decreased range of motion, cubitus valgus deformities, ‘fishtail’ deformities, malunion, non-union, avascular necrosis, and premature epiphyseal fusion.

### 4.1. Lateral Overgrowth

Lateral overgrowth or lateral ‘spurring’ is a hypertrophic bony overgrowth on the lateral side of the elbow due to overstimulation of osteoblasts during the normal bone healing process [47,48]. Lateral overgrowth can be appreciated on conventional radiographs and felt during a physical examination of the elbow [10,29]. The occurrence rate of this complication is comparable between both treatment groups (non-surgical, 4.5–74%; and surgical, 4.5–73.7%) and between surgical techniques (CRPP, 4.5–73.7%; ORIF K-wires, 22.1–73.7%; and ORIF cannulated screw, 10.1–74%) [4,29,34,36,42,49,50,51].

### 4.2. Infections

Infections are an infrequent complication of surgical interventions and can be divided into two groups: superficial surgical site infections and deep infections of the osteosynthesis material. Treatment options for these infections can differ, ranging from local topical (antibiotic) treatment to extensive revision surgery. The occurrence rate of infections as a complication of lateral humeral condyle elbow surgery is 0.01–19.3% [10,36,39,42,43,50,51].

### 4.3. Malunion and Non-Union

A malunion of the bone describes the situation in which a patient’s bone does not heal properly and creates an abnormally shaped joint with possible impaired function of the extremity as a result. A delayed or even non-union of the bone is a failure of a fracture to heal properly after three to nine months [52]. Malunion can cause structural deformities with a cubitus varus or, more commonly, cubitus valgus or impairment in the range of motion. Non-union or malunion of lateral humeral condyle fractures often require revision surgery to attempt to repair shortcomings and improve clinical outcomes for the patient [38,53,54,55,56,57]. Non-union or malunion, as a complication after revision surgery, occurred between 0–13% [54,55,56]. The physical performance score of the elbow, measured using the mayo performance score, increased in more than 80% of patients [54,55,56]. Non-union and malunion of the fracture occur more frequently in the cast therapy group [10,33,42]. The occurrence rate of non-union and malunion of the fracture is between 0–11.8% and 1.3–11.8%, respectively [4,8,10,36,39,40,42,43,49,50,51,53,58]. This rate of occurrence is exceptionally high, demonstrating the unforgiving nature of fracture healing for lateral humeral condyle fractures.

### 4.4. Avascular Necrosis

Avascular necrosis, or osteonecrosis, is a complication that causes ischemic damage to bone cells and, eventually, necrosis of bone due to the loss of blood supply. This can occur to bones after trauma because of the increased swelling, decreased range of motion of the elbow, and rotation of the broken fragment with subsequent tearing of the arteries [40]. The capitulum and lateral condyle are supplied solely with blood from a couple of small end arteries on the lateral side of the elbow. Hence, the capitulum and lateral condyle are considered to be relatively avascular. Therefore, one can appreciate how a slight traumatic injury or surgical operation through a posterior dissection approach can cause permanent damage to the small arteries supplying the lateral side, causing avascular necrosis [59]. The occurrence rate of avascular necrosis after lateral condyle fractures is 0.9–3.1% [8,39,40,49,51,58]. No study has reported the difference in occurrence rates of avascular necrosis between a non-operative and surgical treatment option. The elbow’s functionality, stability, and range of motion are highly impeded after avascular necrosis. As a result, the patient will have a long-term disability when it comes to daily function.

## 5. Conclusions

Lateral humeral condyle fractures are frequently seen in pediatric patients. The Song’s classification is the best validated and most recommended method out of all classification systems developed in the past. Treatment options vary from non-operative treatment with a cast to an open reduction and internal fixation. The success rate for these treatments, i.e., patients with successful fracture healing and without complications, is between 89.8–91.5%. The most notable complications are malunion or non-union, avascular necrosis, postoperative infections, and decreased mobility. Most important, the complications of lateral humeral condyle fractures are quite severe and, if untreated, could lead to a disproportionate loss in the quality of life.

## Figures and Tables

**Figure 1 children-10-01033-f001:**
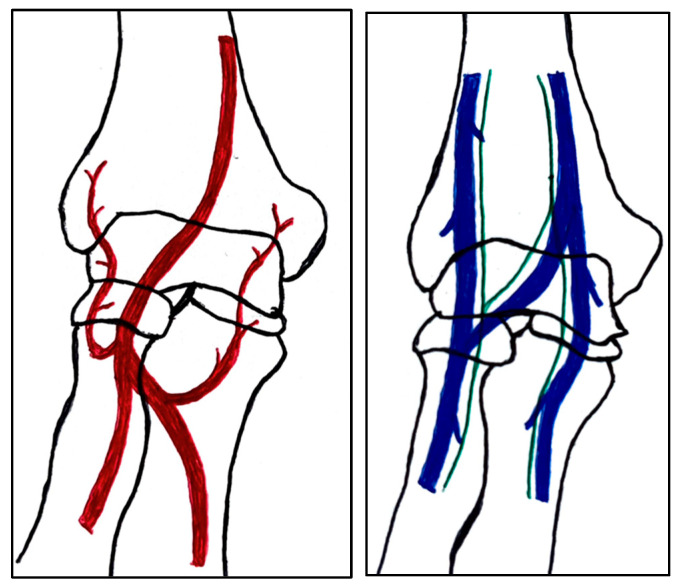
Simple schematic visualization of the anatomy of the arteries (**left**) and the veins with their respective lymphatic vessels (**right**) of the elbow as seen from an anterior view.

**Figure 2 children-10-01033-f002:**
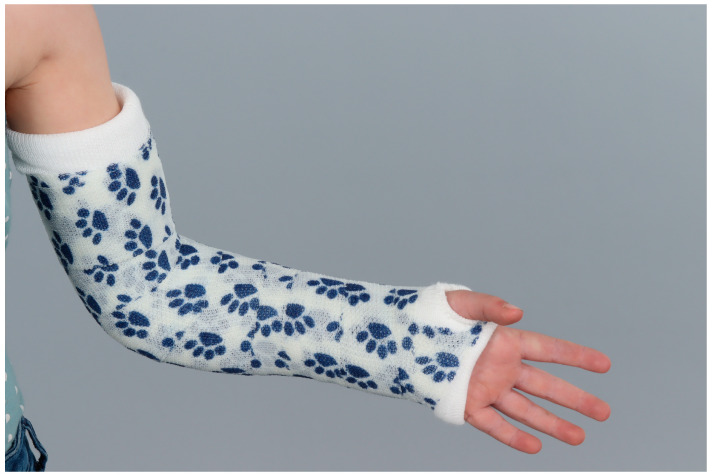
Digital picture of a left arm in an above-elbow cast in the recommended position (90-degree flexion and neutral rotation).

**Figure 3 children-10-01033-f003:**
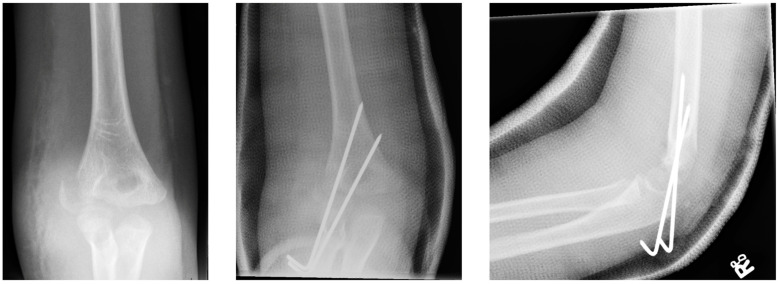
(**Left**) Anterior–posterior view of a lateral humeral condyle fracture. (**Middle**) Anterior–posterior and (**right**) lateral radiographic view of the elbow after open reduction and internal k-wire fixation.

**Table 2 children-10-01033-t002:** Classification of lateral humeral condyle fractures and their respective preferred treatment options for the pediatric patient.

	1 (A)	2 (B)	3 (C)	4	5
Milch [16]	Cast/CRPP	CRPP/ORIF	-	-	-
Jakobs [21]	Cast/CRPP	CRPP	ORIF	-	-
Finnbogason [22]	Cast/CRPP	CRPP	CRPP/ORIF	-	-
Weiss [23]	Cast/CRPP	CRPP	ORIF	-	-
Song [29]	Cast	Cast/CRPP	CRPP	CRPP	ORIF

Cast = above elbow cast, CRPP = closed reduction and percutaneous pinning with Kirschner wires, ORIF = open reduction internal fixation with Kirschner wires and/or cannulated screw.

## Data Availability

No new data were created or analyzed in this study. Data sharing is not applicable to this article.

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
