# Peer review of "Lateral Humeral Condyle Fractures in Pediatric Patients"

_children, 2023, doi:10.3390/children10061033_

Round 1
Reviewer 1 Report
Well written article.
Missing some pictures about the treatment by srcew and possibilities of the resobable implants.
To sum up I suggest to publish it the present form.
Author Response
Thank you for the review
Reviewer 2 Report
Dear authors,
I am pleased to review the submitted paper children-2407620 entitled "Lateral Humeral Condyle Fractures in Pediatric Patients"
The present paper focuses on providing an overview of the epidemiology, anatomy, diagnosis, treatment options, and complications of pediatric lateral humeral condyle fractures based on the most recent literature..
In my opinion the content is original, current, objective and persuasive. But there were some questions need to be answer:
1. Treatment options(Line 171):”Non-operative treatment is the preferred option for fractures with a minor (<2mm) displacement and no other additional injuries[32].”
The treatment did not included the situation for delay diagnosed, for delay diagnosed lateral Humeral Condyle Fractures, is the treatment different from acute fracture?
2. Non-surgical/operative treatment options: Plastered cast therapy(Line 196-197)“If non-operative treatment shows no secondary displacement of the fracture in a long arm cast within the first week after injury, patients should return after 4-6 weeks.”
What is the indication for cast remove when patients return after 4-6 weeks. Here need to be specified.
3. Closed reduction and internal fixation(Line 232-233): “... it is best to visualize a joint reduction through arthrography, arthroscopy, or an arthrotomy.”
It is recommended that this review should cited this following articles:"Li XT, Shen XT, Wu X, Wang S. Ultrasound-assisted closed reduction and percutaneous pinning for displaced and rotated lateral condylar humeral fractures in children. J Shoulder Elbow Surg. 2021 Sep;30(9):2113-2119."
The surgical technique of ultrasound-assisted closed reduction and percutaneous pinning presented in this study can effectively help surgeons reduce displaced and rotated lateral condylar humeral fractures in children to avoid some open reductions and achieve satisfactory outcomes.
4. Closed reduction and internal fixation(Line 239): “Kirschner wires are left in place for 4 weeks after surgery.”
The healing time for Plastered cast therapy is 4-6 weeks, why using K wires only 4 weeks, please add some references.
5. Malunion and non-union(Line 288): “Non-union of lateral humeral condyle fractures often require revision surgery to attempt to repair the shortcoming and improve clinical outcomes for the patient”
Please add some references to describe the prognosis of this revision surgery.
Minor editing of English language required.
Author Response
1. Treatment options(Line 171):”Non-operative treatment is the preferred option for fractures with a minor (<2mm) displacement and no other additional injuries[32].”
The treatment did not included the situation for delay diagnosed, for delay diagnosed lateral Humeral Condyle Fractures, is the treatment different from acute fracture?
> Delayed diagnoses of these fractures, which occurs seldom due to pain and immobility of the child, should be treated as secondary displacement depending on the time between the injury and presentation, and displacement. If displacement is>2mm after 3-7 days after injury, the patient should receive operative treatment. See articles paragraphs 3.1
2. Non-surgical/operative treatment options: Plastered cast therapy(Line 196-197)“If non-operative treatment shows no secondary displacement of the fracture in a long arm cast within the first week after injury, patients should return after 4-6 weeks.”
What is the indication for cast remove when patients return after 4-6 weeks. Here need to be specified.
> Average time for bone to heal in children is between 4-6 weeks. Depending on the severity of the incident, the extensiveness of the fracture, and the clinical presentation of the patient, the physician should choose a treatment length between 4-6 weeks. No references could be found specifying the exact length of casting therapy.
3. Closed reduction and internal fixation(Line 232-233): “... it is best to visualize a joint reduction through arthrography, arthroscopy, or an arthrotomy.”
It is recommended that this review should cited this following articles:"Li XT, Shen XT, Wu X, Wang S. Ultrasound-assisted closed reduction and percutaneous pinning for displaced and rotated lateral condylar humeral fractures in children. J Shoulder Elbow Surg. 2021 Sep;30(9):2113-2119."
The surgical technique of ultrasound-assisted closed reduction and percutaneous pinning presented in this study can effectively help surgeons reduce displaced and rotated lateral condylar humeral fractures in children to avoid some open reductions and achieve satisfactory outcomes.
> Our research team of elbow surgeons still recommend the above. The use of ultrasound is not regular practice and has shown some promising results but not yet decisive enough to out perform arthrography, arthroscopy or arthrotomy.
4. Closed reduction and internal fixation(Line 239): “Kirschner wires are left in place for 4 weeks after surgery.”
The healing time for Plastered cast therapy is 4-6 weeks, why using K wires only 4 weeks, please add some references.
> corrections were made to 4-6 weeks, see comment above
5. Malunion and non-union(Line 288): “Non-union of lateral humeral condyle fractures often require revision surgery to attempt to repair the shortcoming and improve clinical outcomes for the patient”
Please add some references to describe the prognosis of this revision surgery.
> Unfortunately we could not provide enough quality evidence on the true prognosis after revision surgery. We could provide our expert opinion but that is not validated or backed by facts. We hope these results are more available in the future.
Reviewer 3 Report
Dear Authors,
Thank you for the opportunity to review this article.
Introduction is scarce and shoule be exanded.
Line 26-27: What could an unfavourable choice of treatment for undisplaced fractures be? Some authors showed that casting with follow-up X-rays show good results.
Line 22: Laterality of the hand (handedness) is also an important feature that comprises of epidemiology for upper arm / elbow fractures, as shown in this reference here https://www.mdpi.com/2227-9067/8/1/51, and should be noted into Introduction also.
You should establish an aim / objective for your study.
Epidemiology is thorough and is one of the article strengths.
You should include a materials and methods paragraph. What were your criteria of choosing the articles? A flow diagram may be helpful.
Do you have access to postop clinical data? How do the CRPP/ORIF groups present at follow-up? It should be interesting to compare different authors’ choices regarding the same type of fracture in terms of postop.
Minor editing of English language required
Author Response
Line 26-27: What could an unfavourable choice of treatment for undisplaced fractures be? Some authors showed that casting with follow-up X-rays show good results.
> unfavorable choices would be: No cast whilst it was needed or casting when surgery was needed.
Line 22: Laterality of the hand (handedness) is also an important feature that comprises of epidemiology for upper arm / elbow fractures, as shown in this reference here https://www.mdpi.com/2227-9067/8/1/51, and should be noted into Introduction also.
You should establish an aim / objective for your study. Epidemiology is thorough and is one of the article strengths.
> see line 28-30
You should include a materials and methods paragraph. What were your criteria of choosing the articles? A flow diagram may be helpful.
> Our authors selected articles based on the quality of the study, the results presented and the relevance to the review. No extensive systematic literature search has been performed.
Do you have access to postop clinical data? How do the CRPP/ORIF groups present at follow-up? It should be interesting to compare different authors’ choices regarding the same type of fracture in terms of postop.
> We have databases with clinical scores post-op. We would like to use those for a dedicated study on CRPP/ORIF and thought it would divert the purpose of this review towards another goal
Round 2
Reviewer 2 Report
Dear authors,
I am pleased to review the revised paper children-2407620 entitled "Lateral Humeral Condyle Fractures in Pediatric Patients"
The present paper focuses on providing an overview of the epidemiology, anatomy, diagnosis, treatment options, and complications of pediatric lateral humeral condyle fractures based on the most recent literature.
I reviewed this paper before and the authors have now submitted a revised version. I read the authors' response to the reviewer comments, the following questions I raised need to be appropriate answered:
1. Treatment options(Line 171):”Non-operative treatment is the preferred option for fractures with a minor (<2mm) displacement and no other additional injuries[32].”
The treatment did not included the situation for delay diagnosed, for delay diagnosed lateral Humeral Condyle Fractures, is the treatment different from acute fracture?
" Delayed diagnoses of these fractures, which occurs seldom due to pain and immobility of the child, should be treated as secondary displacement depending on the time between the injury and presentation, and displacement. If displacement is>2mm after 3-7 days after injury, the patient should receive operative treatment. See articles paragraphs 3.1"
Not Appropriate. This review aim to providing an overview of the epidemiology, anatomy, diagnosis, treatment options, and complications of pediatric lateral humeral condyle fractures based on the most recent literature, then should include different situation in clinical treatment, but there is no any recommendation of treatment on delay diagnosis over 2 weeks.
2. Non-surgical/operative treatment options: Plastered cast therapy(Line 196-197)“If non-operative treatment shows no secondary displacement of the fracture in a long arm cast within the first week after injury, patients should return after 4-6 weeks.”
What is the indication for cast remove when patients return after 4-6 weeks. Here need to be specified.
"Average time for bone to heal in children is between 4-6 weeks. Depending on the severity of the incident, the extensiveness of the fracture, and the clinical presentation of the patient, the physician should choose a treatment length between 4-6 weeks. No references could be found specifying the exact length of casting therapy."
Not Appropriate. Is there any sign on follow up X ray when patients return? Is the callus critical in patient follow up X ray?
3. Closed reduction and internal fixation(Line 232-233): “... it is best to visualize a joint reduction through arthrography, arthroscopy, or an arthrotomy.”
It is recommended that this review should cited this following articles:"Li XT, Shen XT, Wu X, Wang S. Ultrasound-assisted closed reduction and percutaneous pinning for displaced and rotated lateral condylar humeral fractures in children. J Shoulder Elbow Surg. 2021 Sep;30(9):2113-2119."
The surgical technique of ultrasound-assisted closed reduction and percutaneous pinning presented in this study can effectively help surgeons reduce displaced and rotated lateral condylar humeral fractures in children to avoid some open reductions and achieve satisfactory outcomes.
"Our research team of elbow surgeons still recommend the above. The use of ultrasound is not regular practice and has shown some promising results but not yet decisive enough to out perform arthrography, arthroscopy or arthrotomy. "
Not Appropriate. Again This review aim to providing an overview of the epidemiology, anatomy, diagnosis, treatment options, and complications of pediatric lateral humeral condyle fractures based on the most recent literature, which should widely cover the new concept and treatment. This manuscript may recomend treatment with solid evidence, but should refer all the articles and left them to readers.
4. Closed reduction and internal fixation(Line 239): “Kirschner wires are left in place for 4 weeks after surgery.”
The healing time for Plastered cast therapy is 4-6 weeks, why using K wires only 4 weeks, please add some references.
" corrections were made to 4-6 weeks, see comment above"
Appropriate revised.
5. Malunion and non-union(Line 288): “Non-union of lateral humeral condyle fractures often require revision surgery to attempt to repair the shortcoming and improve clinical outcomes for the patient”
Please add some references to describe the prognosis of this revision surgery.
"Unfortunately we could not provide enough quality evidence on the true prognosis after revision surgery. We could provide our expert opinion but that is not validated or backed by facts. We hope these results are more available in the future. "
Not Appropriate. Please cited following references:
1.Launay F, Leet AI, Jacopin S, Jouve JL, Bollini G, Sponseller PD (2004) Lateral humeral condyle fractures in children: a comparison of two approaches to treatment. J Pediatr Orthop 24(4):385–391
2.Toh S, Tsubo K, Nishikawa S, Inoue S, Nakamura R, Narita S (2002) Osteosynthesis for nonunion of the lateral humeral condyle. Clin Orthop Relat Res 405:230–241
3.Badelon O, Bensahel H, Mazda K, Vie P (1988) Lateral humeral condylar fractures in children: a report of 47 cases. J Pediatr Orthop 8(1):31–34
4.Flynn JC, Richards JF Jr, Saltzman RI (1975) Prevention and treatment of non-union of slightly displaced fractures of the lateral humeral condyle in children. An end-result study. J Bone Jt Surg Am 57(8):1087–1092
5.Flynn JC (1989) Nonunion of slightly displaced fractures of the lateral humeral condyle in children: an update. J Pediatr Orthop 9(6):691–696
6.Masada K, Kawai H, Kawabata H, Masatomi T, Tsuyuguchi Y, Yamamoto K (1990) Osteosynthesis for old, established nonunion of the lateral condyle of the humerus. J Bone Jt Surg Am 72(1):32–40
Minor editing of English language required
Author Response
Dear Reviewer,
We very kindly thank you for the thorough review with extensive feedback. We have made many revisions to the article after inspecting the remarks you have made.
1. Treatment options(Line 171):”Non-operative treatment is the preferred option for fractures with a minor (<2mm) displacement and no other additional injuries[32].” The treatment did not included the situation for delay diagnosed, for delay diagnosed lateral Humeral Condyle Fractures, is the treatment different from acute fracture? This review aim to providing an overview of the epidemiology, anatomy, diagnosis, treatment options, and complications of pediatric lateral humeral condyle fractures based on the most recent literature, then should include different situation in clinical treatment, but there is no any recommendation of treatment on delay diagnosis over 2 weeks.
Our response: Your feedback has led to another deep dive into the available literature on delayed diagnosis and neglected fractures. We have found two good studies presenting treatment options specifically designed for patients with delayed presentation (>3weeks). We have added their recommendations to our text.
The revisions can be found on lines: 329-335, 358-360, 476-478, 498-516
2. Non-surgical/operative treatment options: Plastered cast therapy(Line 196-197)“If non-operative treatment shows no secondary displacement of the fracture in a long arm cast within the first week after injury, patients should return after 4-6 weeks.” What is the indication for cast remove when patients return after 4-6 weeks. Here need to be specified. Is there any sign on follow up X ray when patients return? Is the callus critical in patient follow up X ray?
Our response: We have tried to provide a detailed work flow that a physician can easily follow to determine the successfulness of the treatment and indication to extend cast therapy by using x-ray to detect callus forming in pediatric patients. Revisions can be found on lines: 349-352
3. Closed reduction and internal fixation(Line 232-233): “... it is best to visualize a joint reduction through arthrography, arthroscopy, or an arthrotomy.”
It is recommended that this review should cited this following articles:"Li XT, Shen XT, Wu X, Wang S. Ultrasound-assisted closed reduction and percutaneous pinning for displaced and rotated lateral condylar humeral fractures in children. J Shoulder Elbow Surg. 2021 Sep;30(9):2113-2119."
The surgical technique of ultrasound-assisted closed reduction and percutaneous pinning presented in this study can effectively help surgeons reduce displaced and rotated lateral condylar humeral fractures in children to avoid some open reductions and achieve satisfactory outcomes. This review aim to providing an overview of the epidemiology, anatomy, diagnosis, treatment options, and complications of pediatric lateral humeral condyle fractures based on the most recent literature, which should widely cover the new concept and treatment. This manuscript may recomend treatment with solid evidence, but should refer all the articles and left them to readers.
Our response: We agree that all possibilities should be included in our narrative review to provide the reader with all knowledge so that they, themselves, can decide the preferable option. We have made proper revision to the paragraphs on diagnostic capabilities during closed reduction to include the ultrasound guided surgical options as well. Revisions can be seen on line: 426-435
4. Closed reduction and internal fixation(Line 239): “Kirschner wires are left in place for 4 weeks after surgery.” The healing time for Plastered cast therapy is 4-6 weeks, why using K wires only 4 weeks, please add some references.
Our response: Revisions were made. See lines 346-347
5. Malunion and non-union(Line 288): “Non-union of lateral humeral condyle fractures often require revision surgery to attempt to repair the shortcoming and improve clinical outcomes for the patient”
Please add some references to describe the prognosis of this revision surgery.
Our response: We have read through all the references provided by you and we thank you for excellent literature. We have used a snowball method to find more supporting articles in search for evidence on the prognosis of revision surgery. We have included more information, more evidence and the references provided. Unfortunately most studied we refer to provide subjective evidence: Patient feels better, Range of motion is better, No complications. They do not provide objective results such as performance outcomes score, the amount of complications, and the increase between range of motion before and after surgery. The studies that did provide objective data were included in our review. Revisions were made. See lines 553-555.
We hope to have answered your questions in a proper manner. We look forward to your response and review.
Kind regards,
The Authors
Reviewer 3 Report
Dear Authors,
To summit up, the paper does not bring any new scientific knowledge.
It's neither an systematic review, nor an original article. It is no different then an orthopedic book chapter.
And the comments were not addressed.
Author Response
Dear Reviewer,
We kindly thank you for your time and expertise. We have acknowledged the feedback given from you and made proper revisions to our study. The critical feedback has given more insight to the article.
Line 26-27: What could an unfavourable choice of treatment for undisplaced fractures be? Some authors showed that casting with follow-up X-rays show good results.
Our response: This part of the introduction, for the reader, is quite difficult to understand and does not properly describe the information we want to share. We have made revisions to the introduction to increase readability of the article. These revision can be found on line: 26
Line 22: Laterality of the hand (handedness) is also an important feature that comprises of epidemiology for upper arm / elbow fractures, as shown in this reference here https://www.mdpi.com/2227-9067/8/1/51, and should be noted into Introduction also.
Our response: We have read the article and have included this information in our article for it brings value to the introduction and epidemiology. Revisions can be found on line: 38-39
You should establish an aim / objective for your study.Epidemiology is thorough and is one of the article strengths.You should include a materials and methods paragraph. What were your criteria of choosing the articles? A flow diagram may be helpful.
Our response: We have rewritten a small part of the introduction to better highlight the aim and objective of this study. We thank you for the compliments on our Epidemiology and we have increased the information with the help of your literature suggestions. We think that materials and method sections within this narrative review will not benefit the readability and overall quality of the study. We cannot provide an extensive flowchart with a systematic literature search string because we have not done an extensive systematic review. We have used the most current literature concerning subject matter and searched for additional study with great quality to support or disapprove a specific theory within our narrative review. Revisions can be found on line 28.
Do you have access to postop clinical data? How do the CRPP/ORIF groups present at follow-up? It should be interesting to compare different authors’ choices regarding the same type of fracture in terms of postop.
Our response: Unfortunately, at this time, we don't have database that are readily available to show postop clinical data or CRPP/ORIF comparison at follow-up. However, we think this subject is very interesting since many surgeons still debate about the differences in outcomes between CRPP and ORIF. We have the desire to further investigate these questions, you have asked. We can not provide evidence or data, at this time, to further increase the knowledge within this narrative review.
We hope to have answered your questions in a proper manner. We look forward to your response and review.
Kind regards,
The authors